# Hepatocellular Carcinoma in Patients with Nonalcoholic Fatty Liver Disease: The Prognostic Role of Liver Stiffness Measurement

**DOI:** 10.3390/cancers15030637

**Published:** 2023-01-19

**Authors:** Lucia Cerrito, Irene Mignini, Maria Elena Ainora, Carolina Mosoni, Antonio Gasbarrini, Maria Assunta Zocco

**Affiliations:** CEMAD Digestive Disease Center, Fondazione Policlinico Universitario “A. Gemelli” IRCCS, Università Cattolica del Sacro Cuore, 00168 Rome, Italy

**Keywords:** NAFLD/NASH, hepatocellular carcinoma, ultrasound elastography, magnetic resonance elastography

## Abstract

**Simple Summary:**

Hepatocellular carcinoma (HCC) risk is increased in nonalcoholic fatty liver disease (NAFLD), even without cirrhosis. New noninvasive tests have been proposed to predict HCC risk, and imaging techniques evaluating liver stiffness have gained increasing importance in this context. In this review, we summarize the most recent data about ultrasound and magnetic resonance elastography in estimating HCC risk in patients with NAFLD.

**Abstract:**

Nonalcoholic fatty liver disease (NAFLD), which is nowadays the most common etiology of chronic liver disease, is associated with an increased risk of hepatocellular carcinoma (HCC), with or without cirrhosis. Owing to the high prevalence of NAFLD worldwide, it becomes crucial to develop adequate strategies for surveillance of HCC and new prediction models aiming at stratifying NAFLD population for HCC risk. To this purpose, several noninvasive tests (NITs) have been proposed in the several last years, including clinical parameters, serum biomarkers, and imaging techniques. Most of these tools are focused on the assessment of liver fibrosis. Both ultrasound (US) elastography (especially transient elastography) and magnetic resonance (MR) elastography have been evaluated to estimate HCC risk in NAFLD patients. Recently, the American Association for the Study of Liver Diseases (AASLD) and the European Association for the Study of the Liver (EASL) include these techniques among the recommended NITs for the assessment of liver fibrosis. The aim of this review is to summarize the most recent data on the role of US and MR elastography in HCC risk stratification in patients with NAFLD.

## 1. Introduction

Nonalcoholic fatty liver disease (NAFLD) is a condition of steatosis which arises in the absence of alcohol abuse, viral infection, and other specific causes of hepatic fat overload. Currently, it is estimated that its worldwide prevalence is 32.4% [1]. Moreover, NAFLD incidence and prevalence rates are constantly rising as a consequence of the current spreading of its main risk factors, including hyperlipidemia, obesity, metabolic syndrome, and type II diabetes [2].

Clinical manifestations are heterogeneous and range from isolated hepatic steatosis to nonalcoholic steatohepatitis (NASH), which is characterized by inflammation and liver injury and is strictly associated with progressive liver fibrosis, potentially leading to liver cirrhosis and its complications [3]. NAFLD is also a significant risk factor for the onset of hepatocellular carcinoma (HCC), with rates closely related to the stage of the underlying liver disease. However, although the risk of HCC is higher in patients with cirrhosis, it does not exclusively concern patients with advanced liver disease: a recent Swedish study by Simon et al. reported a progressive increase in HCC incidence rates among patients with simple steatosis, nonfibrotic NASH, fibrosis, and cirrhosis (0.8, 1.2, 2.3, and 6.2 per 1000 person-years, respectively) [4]. A meta-analysis by Younossi et al., including worldwide studies, demonstrated that the risk of HCC increases in NAFLD patients from those with simple steatosis (0.44 per 1000 person-year; range: 0.29–0.66) to those with NASH (5.29 per 1000 person-years; range: 0.75–37.56) [2]. Moreover, Piscaglia et al., in their multicenter observational prospective study on a population of 145 patients with NAFLD, underline the importance of regular ultrasound surveillance even in patients with simple steatosis in order to detect HCC at an earlier stage, since 46.2% of NAFLD-associated HCC occurred in noncirrhotic subjects [5]. Similar results were obtained by other studies, describing a 41.7–49% HCC development rate in NAFLD patients without a cirrhotic background [6,7]. The risk of HCC in NAFLD-related cirrhosis seems to be lower than that observed in hepatitis C virus (HCV)-related cirrhosis [8], but, differently to HCV, NAFLD increases the risk of HCC even without cirrhosis [9].

The enormous increase in worldwide incidence of metabolic syndrome, together with the progress in the therapeutic management of viral hepatitis, sets NAFLD among the fastest rising causes of HCC [10,11,12]. Metabolic dysfunction has gained a leading role in fatty liver disease pathogenesis and a recent consensus by experts has postulated a new inclusive definition to better highlight this new perspective: MAFLD (metabolic associated fatty liver disease) [13]. Consistently, together with NALFD, MAFLD is also recognized as an emerging cause of HCC [14]. In a recent paper based on a database by the ITA.LI.CA (Italian Liver Cancer) group, Vitale et al. analyzed 6882 patients aiming to outline epidemiological trends of MAFLD-related HCC and found that the prevalence of HCC in both single etiology MALFD and mixed etiology MALFD (metabolic and others) is rapidly increasing over time [15]. The burden of this phenomenon and the consequent growth of healthcare costs are substantial, and they are expected to further increase in the coming years [11].

The onset of HCC in NAFLD patients is the result of the interaction of ethnicity, genetics, comorbidities, gut microbiome, and environment. Among these underlying factors, immune and inflammatory responses, oxidative stress, deoxyribonucleic acid (DNA) damage, and autophagy take part in carcinogenesis in a complex mutual interplay [16,17]. In such a variegated scenario, liver fibrosis seems to play a key role: it is a dynamic process that arises in response to chronic liver injury by the accumulation of extracellular matrix, with subsequent development of fibrotic scars. This process can progressively impair liver function both by deeply affecting its mechanical properties and by altering the intercellular signal pathways [18]. The onset and the stage of liver fibrosis appear to be both related with overall and disease-specific mortality in NAFLD patients; therefore, fibrosis is currently recognized as the main predictive factor of long-term outcome of NAFLD [19]. To assess and stage liver fibrosis is consequently crucial to predict the risk of cirrhosis and HCC and to build adequate surveillance strategies both in cirrhotic and noncirrhotic NAFLD patients.

Liver biopsy is the gold standard for the staging of liver fibrosis, but it is an expensive, invasive procedure and, although rarely, can be complicated by clinically serious events including bleeding, infection, pneumothorax, hemothorax, and death [20]: for these reasons, it does not represent a suitable diagnostic method for such a prevalent condition.

In the last several years, a significant effort to identify noninvasive tests (NITs) to stage liver fibrosis has been made, with great progress in the field of both serological parameters and imaging techniques.

Serum biomarkers represent a safer and lower cost alternative to liver biopsy and are gaining widespread use. However, although they commonly show acceptable negative predictive value, they are often encumbered with a high rate of false positive results. Both direct and indirect markers of liver fibrosis can be assessed. The first group includes markers which directly measure extracellular matrix degradation and fibrogenesis, for example, amino-terminal pro-peptide of type III collagen (Pro-C3), hyaluronic acid, and tissue inhibitor of metalloproteinase-1 (TIMP-1). Conversely, indirect markers are the serological tests commonly used to assess liver damage and its complications, such as portal hypertension [21]. Many serum tests incorporate direct and indirect markers of fibrosis, also considering comorbidities in some cases. Some of them were designed specifically for NAFLD, such as NAFLD fibrosis score (NFS) and the BARD score [22]. Other tests, including Fibrosis 4 (FIB-4), aspartate aminotransferase (AST) to platelet ratio index (APRI), and enhanced liver fibrosis score (ELF), were originally studied for HCV, but can be used to assess any chronic liver disease [21].

Among imaging techniques, elastography has gradually spread in clinical practice to diagnose and stage fibrosis in patients with chronic liver disease, through the assessment of hepatic tissue stiffness by the analysis of its mechanical properties: increased stiffness is an index pointing out the presence of fibrosis [21].

Both ultrasound (US) and magnetic resonance (MR) based techniques can be used to measure liver stiffness (LS) noninvasively. The points of strength and weakness of the aforementioned methods to predict HCC risk in NAFLD are summarized in Table 1.

The present review explores the prognostic role of liver stiffness measurement with ultrasound and magnetic resonance elastography (MRE) for assessing the risk of HCC in patients with NAFLD.

## 2. Liver Stiffness Assessment

The American Association for the Study of Liver Diseases (AASLD) highlights the importance to detect advanced fibrosis in NAFLD through NITs, which could reliably replace liver biopsy in the diagnosis and management of NAFLD [25]. More recently, the European Association for the Study of the Liver (EASL) published an update of its previous guidelines on NITs for the evaluation of liver disease severity and prognosis, stressing how this topic has gained increasing interest in the last several years [26].

Imaging techniques that can assess noninvasively the physical properties of liver tissue appear promising in this context. Liver stiffness measurement (LSM) represents the milestone among these imaging parameters and can be obtained by different methods: vibration-controlled transient elastography (TE—FibroScan^®^), shear wave elastography (SWE), and MRE.

TE is a one-dimensional US technique that measures the velocity of low-frequency shear waves propagating through the liver, which is directly related to LS. It is the most widely applied method and it is fast, easy to learn, largely available, and provides good reproducibility [27].

SWE includes point SWE (pSWE), also known as acoustic radiation force impulse imaging (ARFI), and two-dimensional SWE (2D-SWE). Similarly to TE, pSWE provides a one-dimensional measure of tissue elasticity. It is based on short-duration acoustic pulses that propagate through the tissue and generate shear waves whose velocity is measured in a specific region of interest (ROI). Despite using the same pSWE technique, 2D-SWE produces mechanical excitation in multiple focal zones of the liver, inducing lateral shear waves, and provides real-time acquisition of shear wave velocity [28,29].

MRE uses special software and hardware to generate mechanical waves. Quantitative images of the liver can be rapidly obtained during breath-hold acquisitions and they allow calculation of regional mechanical properties through specific formulas [26,30].

The AASLD recommends both TE and MRE as equally useful modalities to evaluate advanced fibrosis. While MRE shows a better performance to identify low-grade fibrosis, TE is quicker and lower cost and has a comparable accuracy in the case of advanced fibrosis [25]. Owing to its wide availability and low cost, EASL recommends TE as the first-line imaging method, identifying a cutoff of 8 kiloPascal (kPa) as the most validated threshold to rule-out advanced fibrosis in NAFLD [26].

### 2.1. Ultrasound-Based Assessment

Among ultrasonographic techniques for LS assessment in NAFLD, current literature focuses especially on TE. In the last several years, different studies evaluated the association between HCC risk and TE, alone or in combination with clinical or serological biomarkers, and some risk prediction scores based on ultrasound have been reported.

Shili-Masmoudi et al. published an observational prospective study on 2251 patients affected by NAFLD of any degree of severity, in which the diagnosis of NAFLD was based on the presence of liver steatosis at US. LS was measured by FibroScan^®^ at baseline and data about overall survival (OS), cardiovascular and liver-related events, including HCC, were collected during the follow-up period. They stated the role of LSM as an independent prognostic factor in the prediction of OS (in both univariable and multivariable Cox proportional hazard models) and liver complications: they were more common in patients with elevated LSM. Moreover, the risk of HCC increased according to baseline LSM: HCC incidence was 0.32% for LSM < 12 kPa, 0.58% for LSM between 12 and 18 kPa, 9.26% for LSM between 18 and 38 kPa, and 13.3% for LSM > 38 kPa [31].

Similar results were obtained by Petta et al., who performed LSM by FibroScan^®^ in a cohort of 1039 patients with NAFLD-related compensated advanced chronic liver disease (cACLD) at baseline and after 1 year. Based on the difference between follow-up and baseline LSM (Δ-LSM), three groups of patients were identified: improved (reduction of more than 20%), stable (from 20% reduction to 20% increase), and impaired LS (an increase of more than 20%). Both baseline LSM and the Δ-LSM were prognostic factors for liver decompensation and Δ-LSM significantly predicted HCC occurrence rates: 2.4% in patients with improved LSM, 3.4% in the case of stable LSM, and 6.7% in the case of impaired LSM. Therefore, they proposed an algorithm to stratify the risk of HCC, liver decompensation, and death using baseline LSM and Δ-LSM: HCC risk was modest for improved LS, intermediate for stable LS, and high for impaired LSM. They pointed out LSM as an independent variable of Cox regression related to HCC development (HR, 1.03; 95% confidence interval—CI 1.00–1.04; *p* = 0.003), together with female sex and age. In particular, the authors found that Δ-LSM was a significant predictor of HCC occurrence compared to baseline LSM, (HR: 1.72; 95% CI, 1.01–3.02; *p* = 0.04 and HR: 1.02; 95% CI 0.98–1.05; *p* = 0.27, respectively). Even if this result requires further validation on a larger cohort, it could represent an important tool for the individualization of follow-up and prognosis assessment in subjects with NAFLD and cACLD [32].

Another interesting US tool for the evaluation of patients with NAFLD is the controlled attenuation parameter (CAP), which was developed to quantitatively estimate the amount of hepatic fat. In particular, it quantifies the attenuation of ultrasonic waves as they propagate through the liver and it is expressed in decibels per meter (dB/m). It is measured by Fibroscan^®^ simultaneously with LS and in the same region of interest [33]. 

Similar to LS, the role of CAP in the prediction of HCC risk has been explored in recent studies, but the results are controversial.

Liu et al. evaluated 4284 patients with chronic liver disease, including 1542 with NAFLD. They all underwent LSM and CAP measurement by TE and, during the follow-up period, 45 subjects presented liver-related events (34 patients had HCC). The multivariate analysis, performed both on the whole population and on the NAFLD-subgroup, reported that LSM independently predicted liver-related events (including HCC) together with low platelets count, low albumin serum levels, male sex, and diagnosis of chronic HBV infection. On the other hand, elevated CAP values (≥248 dB/m) did not efficiently predict the occurrence of liver-related events on multivariate analysis, even if CAP values appeared to be protective on univariate analysis (HR 0.485 with 95% CI 0.240–0.980; *p* = 0.044) and when considered as a continuous variable (HR 0.995 with 95% CI 0.990–1.000; *p* = 0.068). Unfortunately, the authors considered HCC as part of liver-related complications and did not perform a separate analysis of its occurrence in relation to LSM [34].

In another study by Izumi et al., the combination of LS and CAP measurements were useful for HCC screening and risk stratification both in viral-related chronic liver disease and NAFLD. They enrolled 1054 patients (258 with NAFLD) and calculated LS and CAP cut-offs for each etiology. Surprisingly, they observed that HCC incidence in the NAFLD-subgroup was higher for patients with LS ≥ 5.4 kPa and CAP ≤ 265 dB/m (HR 8.91, 95% CI 1.47–67.97, *p* = 0.0192) [35].

More recently, Braude et al. published a multicenter prospective study on the association between VCTE and overall mortality in a large Australian population of 22,653 NAFLD patients undergoing VCTE (including LS and CAP measurements) between 2008 and 2019. Two-hundred and seventeen deaths were registered during the follow-up and HCC occurred in thirteen patients (6%). For multivariate analysis, the authors observed that an increase in LS was associated with a growth in overall mortality, including HCC (HR 1.02 per each kPa of increase, CI 1.01–1.03, *p* < 0.001) and, in particular, the cutoff of 10 kPa was significantly related to mortality (HR 2.31, CI 1.73–3.09, *p* < 0.001). On the other hand, CAP values were not associated with the risk of death based on univariate analysis (HR 1.00, CI 1.0–1.0, *p* = 0.488). This study underlines the role of LSM (measured through VCTE), together with age and Charlson comorbidity index, as a predictive factor for all-cause mortality (HCC included) [36].

Sugihara et al. performed a cross-sectional study on 181 Japanese patients and analyzed the combination of LSM and CAP to identify the subjects with chronic liver diseases (NAFLD/NASH, HBV, HCV) at high-risk to develop HCC. LSM alone (cutoff 5.3 kPa) had 100% sensitivity and 75% specificity in HCC detection with an area under the receiver operating characteristic curve (AUROC) of 0.88. The association of LSM and CAP proved its effectiveness in identifying patients at high risk for HCC, with 90% sensitivity and 99% negative predictive value (*p* = 0.006): the cutoff was 5.3 kPa for LSM with any value of CAP and 248 dB/m for CAP with any value of LSM. Despite these interesting results highlighting the advantages of LSM and CAP as potential instruments in population screening, the value of the study is quite limited by its small size and absence of multivariate analysis [37].

Nevertheless, TE may be useful to stratify the risk of HCC in NAFLD and to select patients who deserve stricter medical follow-up. In the last several years, different risk models combining LSM and/or CAP with nonimaging biomarkers (clinical or serological parameters) have been proposed. 

Lee et al. developed a predictive model analyzing 2666 NAFLD patients in the training cohort and 467 patients in the validation cohort with a median follow-up of 64.6 months. The diagnosis of NAFLD was established with CAP measurements obtained by TE and during the follow-up 22 patients (0.8%) of the training set developed HCC. Through a multivariate analysis the authors constructed a predictive model for HCC development including the following variables: age ≥ 60 years (HR 9.1), aspartate aminotransferase (AST) ≥ 34 IU/L, platelet count <150 × 10^3^/μL (HR 3.7), and LS ≥ 9.3 kPa (HR 13.8). The AUROC for HCC prediction was 0.948, 0.947, 0.939 at 2-, 3-, 5-years, respectively, in the training cohort, and 0.777, 0.781, 0.784 at 2-, 3-, 5-years, respectively, in the validation cohort [38].

Miura et al. performed a cross-sectional study on 191 NAFLD subjects with comorbidities to investigate the role of different US-based NAFLD scoring systems in narrowing the group at high risk of HCC. For this purpose, they stratified the patients in three groups (Low-risk, intermediate-risk and high-risk) by Fibrosis-4 score, based on age, AST, ALT, and platelet count. In the first group no HCC was detected; NAFLD scoring systems were calculated only for the two latter groups (intermediate and high risk), thus narrowed the analysis to 120 patients. The most useful VCTE-based scores were Agile 3+ and Agile 4+, designed to detect subjects with NAFLD and advanced fibrosis. They share the majority of the variables: Agile 3+ considers age, AST/alanine aminotransferase (ALT) ratio, platelets count, diabetes status, and LSM, while in Agile 4+ age is excluded. After the first modest but useful selection obtained by Fibrosis-4 index, Agile 3+ proved to be useful in further narrowing the HCC-risk group (high sensitivity, high negative predictive value): 26 HCC among 80 patients (33%). However, this was a single-center study with a small sample size; thus, the results should be validated in larger prospective studies [39].

### 2.2. Magnetic Resonance Assessment

Several studies compared the diagnostic performance of MRE compared to TE in quantifying both steatosis and fibrosis in NAFLD patients and demonstrated that MRE and MR-based proton density fat fraction (PDFF) are more precise than TE and CAP measurements in the assessment of all grades of liver fibrosis [40,41,42]. These techniques are not affected by factors such as body mass index (BMI), ascites, bowel gas, or operator experience, and they can estimate the stiffness of the entire liver, whereas TE examines only selected areas [43]. They can also quantify hepatic fat accumulation using PDFF that shows a better accuracy than CAP in assessing all grades of steatosis. However, MRE is more expensive, time-consuming, and less widely available than US-TE: for these reasons, cost-effectiveness studies will be necessary to define the most appropriate screening strategy for NAFLD.

A systematic review and pooled analysis by Hsu et al. on 230 patients with biopsy-diagnosed NAFLD confirmed the superiority of MRE compared to TE in identifying the different stages of fibrosis in NAFLD, and provided optimal MRE thresholds to distinguish the different degrees of fibrosis: the AUROC of MRE versus TE was 0.82 (95% CI, 0.76–0.88) vs. 0.87 (95% CI, 0.82–0.91) (*p* = 0.04) for stage-1, 0.87 (95% CI, 0.82–0.91) vs. 0.92 (95% CI, 0.88– 0.96) (*p* = 0.03) for stage-2, 0.84 (95% CI, 0.78–0.90) vs. 0.93 (95% CI, 0.89–0.96) (*p* = 0.001) for stage-3, 0.84 (95% CI, 0.73–0.94) vs. 0.94 (95% CI, 0.89–0.99) (*p* = 0.005) for stage-4 [44].

Only a few studies evaluated the role of MRE in predicting liver complications and especially HCC in NAFLD patients. Anaparthy et al. compared LS by MRE in 30 compensated cirrhotic patients with HCC (cases) and in 60 patients without HCC (controls). The etiologies of underlying liver disease were heterogeneous: 10% alcoholic, 33% NAFLD, and 57% viral. Despite the findings of several previous US studies, no significant associations were identified between MRE-LS and HCC development in compensated cirrhosis; in fact, similar LS values were assessed in liver areas far from the neoplastic lesion in both cases (6.1 ± 2.0 kPa) and controls (6.3 ± 2.5 kPa; *p* = 0.7). The researchers hypothesize that these discrepancies could be determined by a LS overestimation obtained during US measurement by the influence of peritumoral oedema and liver inflammation (expressed by transaminases elevation). Furthermore, they admitted that the small sample size does not allow for drawing definitive conclusions [45].

More recently, a retrospective study published by Ichikawa et al. aimed to stratify HCC risk by MRE in chronic liver disease. They enrolled 161 patients stratified in three groups according to the differences in MRE-measured LS values obtained after two different examinations at an interval of about 12 months: group A, high values on first MRE (60 patients); group C, low values on both MRE (36 patients); group B, other combinations (65 patients). During the follow-up period, HCC was diagnosed in 47 subjects (29.2%) divided as follows: 46.7% in group A (28/60), 26.2% in group B (17/65), 5.6% in group C (2/36). After a 3-year follow-up, a significant difference in the rate of HCC development was demonstrated among the different groups (A: 45.1%, B: 26.1%, C: 12.4%; *p* = 0.0002). Thus, the authors underlined the importance of MRE-LS stratification for HCC risk assessment in chronic liver disease together with age and ALT levels (risk ratio 1.018–6.030; *p* = 0.0028–0.0268) [46].

Furthermore, Higuchi et al. retrospectively evaluated the association between MRE and liver-related complications (HCC, functional decompensation, extrahepatic tumors, major cardiovascular events, death) in a large cohort of 2373 subjects with chronic liver disease (20.5% patients with NAFLD) and identified a direct correlation between HCC risk and LS increase. In particular, for each 1 kPa increase in LS the adjusted hazard ratios (HR) (with 95% CI) for HCC, decompensation, extrahepatic tumors, major cardiovascular events, and mortality were 1.28 (1.2–1.4), 1.34 (1.3–1.4), 1.00 (0.9–1.1), 0.96 (0.9–1.1), and 1.17 (1.1–1.2), respectively. In the case of established cirrhosis, the adjusted HR (with 95% CI) for HCC, decompensation, extrahepatic tumors, major cardiovascular events, and mortality were 4.20 (2.2–8.2), 67.5 (9.2–492), 0.90 (0.5–1.7), 0.83 (0.4–1.7), and 2.90 (1.6–5.4), respectively [47]. Unfortunately, these findings were referred to patients with chronic liver disease caused by different etiologies and should be confirmed in selected NAFLD cohorts.

A recent pooled meta-analysis by Ajmera et al., performed through a systematic search in the cohorts of six previous studies, underlined the prognostic role of LS measured with MRE for the assessment of liver-related events, including HCC, in 2018 patients with NAFLD (median baseline MRE 4.15 kPa) and a median 3-year follow-up. The primary outcome was a global endpoint comprehensive of ascites, hepatic encephalopathy, esophageal varices needing treatment, HCC, and death with a HR of 11.0 (95% CI: 7.03–17.1, *p* < 0.001) for MRE 5–8 kPa and 15.9 (95% CI: 9.32–27.2, *p* < 0.001) for MRE > 8 kPa. The patients included in the analysis had a positive MEFIB index (derived from the association of FIB-4 ≥ 1.6 kPa and MRE ≥ 3.3 kPa) with stage-2 fibrosis or superior. The authors observed that this score had a HR of 20.6 for the primary outcome (95% CI: 10.4–40.8, *p* < 0.001), while the predictive value was negative (99.1% at 5 years) with a negative MEFIB index, thus demonstrating its role as strong negative predictor for liver decompensation. At the same time, the risk to develop HCC after 3 years was 5.66% with MRE ≥ 8 kPa, 5.25% with MRE 5–8 kPa, and 0.35% with MRE < 5 kPa [48].

## 3. Discussion

Since cirrhosis is the cause of the large majority of HCC, surveillance for HCC in cirrhotic patients is well defined by international guidelines. B-mode US is the method of choice and a six-month interval is currently recommended [49,50]. However, some other conditions, notably NAFLD or chronic HBV infection, may lead to HCC onset even in noncirrhotic livers. In these cases, the precise estimation of HCC risk is challenging in terms of both timing and cost-effectiveness. In 2012, White et al. published the first systematic review of epidemiologic data on HCC in NAFLD. They included 17 cohort studies, 18 case-control studies, and 26 case series and found that HCC risk was much higher in HCV-related disease than NAFLD and markedly increased in cirrhotic livers [51]. Even if HCC cumulative risk over two decades was lower than 3% in NAFLD/NASH [51], the actual growing prevalence of this condition worldwide raises particular concerns. Therefore, the urgent need for reliable NITs for predicting HCC in such individuals is widely recognized. HCC incidence rates in noncirrhotic patients deserve specific surveillance policies and it is essential to identify subjects who should be more strictly monitored.

In such a setting, the rapid appearance of new cheap and easily reproducible serological biomarkers arouses an undoubted interest, but they still have suboptimal performances. As liver fibrosis represents the main prognostic factor for HCC, the predictive role of imaging techniques assessing LS has become a new research issue.

The first findings about the association of LS and HCC development concern viral chronic liver diseases: in 2008 a cross-sectional study by Masuzaki et al. highlighted the significant role of LSM by TE in HCC prediction for patients with chronic HCV-infection [52], and in 2011 Jung et al. proved the usefulness of LSM for predicting HCC in subjects with chronic HBV-infection [53].

LS-based prediction paradigms have also been proposed: Shin et al. designed a model for HCC prediction in HBV patients considering LS, spleen diameter, and platelet count [54], and Liu et al. reported the role of C-reactive protein in improving LS prognostic role for HCC onset in HBV-related cirrhosis [55]. An association between the reduction in LS during treatment and decreased rates in HCC occurrence has been described in chronic HBV [56]. When considering HCV patients who had undergone successful virus eradication, different studies show that both pre- and post-treatment LSM are useful for predicting HCC risk after viral clearance [57,58,59].

Data concerning NAFLD flourished recently and they are convergent with that reported for chronic viral infections. In particular, current literature highlights the prognostic role of TE and, less frequently, of MRE. Similar to what was observed in patients with viral hepatitis, LSM proved to be strongly associated with HCC risk in all the examined NAFLD/NASH cohorts. No comparative studies between NAFLD and non-NAFLD etiologies are yet available on this topic, and current data are too heterogeneous in terms of population, outcomes, and imaging parameters analyzed to allow an assessment of specific cutoffs of LSM for different etiologies. Nonetheless, considering NAFLD growing incidence and the risk of HCC even in the absence of cirrhosis, more widespread use of LSM assessment in NAFLD patients compared with other chronic liver disease etiologies may be hypothesized. Especially US-based techniques, owing to their availability and cost-effectiveness, may have a role in screening programs for HCC in selected groups. Indeed, current data may be insufficient to narrow the risk group in such a wide population. Therefore, as for chronic viral liver diseases, also for NAFLD patients, an effort has been made to find new composite scores, combining laboratory and imaging parameters (Table 2) for this purpose. Among them, Agile 3+ and Agile 4+ indexes, based on a stepwise approach, now seem to be the best pathways to categorize patients and identify those who should undergo more in-depth assessments.

## 4. Conclusions and Future Perspectives

Actual data about the role of NITs in predicting HCC risk in NAFLD patients are promising, but still far from being exhaustive. LSM stands out as the most analyzed tool and has proved to be a reliable prognostic factor for liver fibrosis and HCC, both alone or in combination with different imaging and/or serological markers. The role of other US parameters, such as CAP, remains controversial and should be matter of further investigation. Among the various imaging methods, US-based elastography seems to be more suitable for surveillance programs than MRE, which is more expensive and time-consuming, despite its better accuracy. However, current literature focuses only on the association between TE and HCC, and no data about other US techniques (namely pSWE and 2D-SWE) are so far available. New prospective studies are needed to clarify their role in HCC prediction. Moreover, LSM by TE is encumbered by some limitations. It requires a dedicated device, and it is operator-dependent and affected by different factors such as obesity, presence of intestinal gas, ascites, food intake, or acute hepatitis. Owing to the large spread in some of the aforementioned conditions in NAFLD patients, studies analyzing these specific populations in order to identify the most accurate and cost-effective instrument to predict HCC should be encouraged.

## Figures and Tables

**Table 1 cancers-15-00637-t001:** Advantages and disadvantages of the currently available techniques for predicting HCC risk in NAFLD.

	Advantages	Disadvantages
Serum test-based scores (NAFLD fibrosis score, BARD score) [22]	- Easily accessible - Safe - Low cost - Acceptable negative predictive value	- High rates of false positive results
Ultrasound-based elastography [23,24]	- Widespread - Good reproducibility - Suitable for surveillance programs	- Operator-dependent - Affected by food intake, ascites, bowel gas, obesity and acute hepatitis
Magnetic resonance elastography [23,24]	- Accurate - Not influenced by BMI, ascites, bowel gas, operator’s experience	- Expensive - Time-consuming - Hardly available

**Table 2 cancers-15-00637-t002:** The role of ultrasound liver stiffness measurement in predicting HCC risk in NAFLD patients.

Author (year)	Study Population (n)	LSM and HCC Risk
Sugihara (2017) [37]	181 (NAFLD/NASH, HBV, HCV)	LSM alone (cutoff 5.3 kPa): HCC detection → 100% sensitivity, 75% specificity, 0.88 AUROC
	HCC: 3	Association LSM+CAP (LSM cutoff 5.3 kPa with any CAP; CAP > 248 dB/m with any LSM): effectiveness in identifying subjects with liver disease at high risk for HCC → 90% sensitivity, 55% specificity, 10% positive predictive value, 99% negative predictive value; *p* = 0.006
Liu (2017) [34]	4284 (1542 NAFLD of any degree)	LSM independently predicted liver-related events (including HCC)
	HCC: 34	CAP ≥ 248 dB/m: no significant predictor for HCC occurrence in multivariate analysis (on univariate analysis: HR 0.485 with 95% CI 0.240–0.980; *p* = 0.044; continuous variable: HR 0.995 with 95% CI 0.990–1.000; *p* = 0.068)
Izumi (2019) [35]	1054 (258 NAFLD of any degree)HCC: 88	Higher HCC incidence (in NAFLD subgroup): - LS ≥ 5.4 kPa - CAP ≤ 265 dB/m (HR 8.91, 95% CI 1.47–67.97, *p* = 0.0192)
Shili-Masmoudi (2020) [31]	2251 NAFLD of any degree	LSM < 12 kPa: HCC incidence → 0.32% LSM between 12 and 18 kPa: HCC incidence → 0.58% LSM between 18 and 38 kPa: HCC incidence → 9.26% LSM > 38 kPa: HCC incidence → 13.3%
Lee (2021) [38]	NAFLD of any degree: - 2666: training cohort - 467: validation cohort HCC: 22 in training cohort	LS ≥9.3 kPa (HR 13.8) in a multivariate analysis with non-US variables: age ≥60 years (HR 9.1), AST > 34 IU/L, platelet count <150 × 10^3^/μL (HR 3.7), LS ≥9.3 kPa (HR 13.8).AUC for HCC prediction: - Training cohort: 0.948 (at 2 years), 0.947 (at 3 years), 0.939 (at 5 years) - Validation cohort: 0.777 (at 2 years), 0.781 (at 3 years), 0.784 (at 5 years)
Petta (2021) [32]	1039 NAFLD with cACLD	Δ-LSM significant predictor of HCC occurrence (HR: 1.72; 95% CI, 1.01–3.02; *p* = 0.04): - Improved LSM (Δ-LSM < 20%) → low HCC risk - Stable LSM (Δ-LSM −20% to +20%) → intermediate HCC risk - Impaired LSM (Δ-LSM < 20%) → high HCC risk
Braude (2022) [36]	22653 NAFLD HCC: 13	LS increase: growth in overall mortality (including HCC: HR 1.02 per kPa, CI 1.01–1.03, *p* < 0.001) VCTE-LSM >10 kPa: related with mortality (HR 2.31, CI 1.73–3.09, *p* < 0.001) CAP: not associated with risk of death in univariate analysis (HR 1.00, CI 1.0–1.0, *p* = 0.488)

cACLD = chronic advanced compensated liver disease; CAP = controlled attenuation parameter; LSM = liver stiffness measurement; Δ-LSM = difference between follow-up and baseline LSM; ALT = alanine aminotransferase; AST = aspartate aminotransferase; NAFLD = nonalcoholic fatty liver disease.

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
