# Peer review of "Hepatocellular Carcinoma in Patients with Nonalcoholic Fatty Liver Disease: The Prognostic Role of Liver Stiffness Measurement"

_cancers, 2023, doi:10.3390/cancers15030637_

Round 1

Reviewer 1 Report

The manuscript by Cerrito et al. constitutes a comprehensive review on the prognostic role of liver stiffness measurement in the risk prediction and diagnosis of HCC in patients with NAFLD. The authors provide here a very well-written and complete revision on the topic, focusing on the usefulness of distinct non-invasive methods to evaluate liver stiffness and fibrosis in the prediction of the risk for HCC development in patients with NAFLD. The revision is well-structure and I only have small concerns that might be addressed before final acceptance for publication. 

-       Please carefully revise the manuscript since there a lot of typos and spaces lacking between words. 

-       Et al should be in italics

-       It would be interesting to discuss a little bit if the presented approaches are differently used considering the etiology of HCC (NAFLD vs non-NAFLD). Are there differences in cut-offs, sensitivity, specificity, etc? If no studies are available on this regard, the authors may want to provide their critical vision and opinion in this matter. 

Author Response

  • Please carefully revise the manuscript since there a lot of typos and spaces lacking between words

Re: Thank you for your observation. Typos and lacking spaces have been corrected.

  • Et al should be in italics

Re: “Et al” has been converted in “et al” using italics.

  • It would be interesting to discuss a little bit if the presented approaches are differently used considering the etiology of HCC (NAFLD vs non-NAFLD). Are there differences in cut-offs, sensitivity, specificity, etc? If no studies are available on this regard, the authors may want to provide their critical vision and opinion in this matter.

Re: following the suggestion, discussion has been implemented in order to include authors’ opinion on the different uses of the presented imaging techniques according on HCC aetiology.

Reviewer 2 Report

Both US elastography and MR elastography have been evaluated to stratify HCC risk in NAFLD patients. This review characterizes each elastography and tackles the key publications. Although relevant, published studies are generally less in NAFLD than in viral etiologies, risk stratification using baseline, follow-up and changes in stiffness values alone or in combination are addressed in the current review.

"Simply" summary?
Why capatilized Nonalcoholic Fatty Liver Disease (NAFLD)? Magnetic Resonance (MR)?
Line 306: "withchronic liver disease"?
Line 374:  "Howevercurrent"

Author Response

  • "Simply" summary?

Re: we removed the word “simply”, leaving only “summary”.

  • Why capatilized Nonalcoholic Fatty Liver Disease (NAFLD)? Magnetic Resonance (MR)?

Re: Thank you for your observation. We removed capital letters.

  • Line 306: "withchronic liver disease"?

Re: lacking space was added

  • Line 374: "Howevercurrent"

Re: lacking space was added

Reviewer 3 Report

The role of liver stiffness measurement (LSM) in NAFLD to predict HCC risk is a timely topic warranting a comprehensive review.  

The review by Lucia et al. is very interesting because it originally discusses quite exhaustively the role of non-invasive LSM using the TE and MRE at the interface between NAFLD, a pandemic hitting 1/3 of the overall population, and the development of HCC.

I sincerely recommend the publication of this review, although some minor points should be addressed for the improvement of the paper.

Points to be addressed:

-Introduction. Authors focused on recent increasing trend of NAFLD associated HCC. However, it will be also interesting to describe trend on the rise of MAFLD (a consensus driven proposed nomenclature for metabolic associated fatty liver disease. Gastroenterology 2020;158:1999-2014.e1.) associated HCC. 

- Pages 3. Table 1. Although table 1 is very adequate, I miss some example studies. Important study examples will help the reader's understanding.

- An important systemic review highlighting the HCC risk in NAFLD (Clin Gastroenterol Hepatol. 2012;10:1342-1359.e2.) should be cited in ‘Discussion’.

- There are some minor typographical errors that should be corrected (abbreviations, points, spaces, grammar-related, mellitus in cursive …).

Author Response

  • Authors focused on recent increasing trend of NAFLD associated HCC. However, it will be also interesting to describe trend on the rise of MAFLD (a consensus driven proposed nomenclature for metabolic associated fatty liver disease. Gastroenterology 2020;158:1999-2014.e1.) associated HCC.

Re: thank you for your useful suggestion, which perfectly fits our paper. We included in the introduction recent data about HCC epidemiology in the new landscape of MALFD

  • Pages 3. Table 1. Although table 1 is very adequate, I miss some example studies. Important study examples will help the reader's understanding.

Re: specific references have been introduced to support table 1 content

  • An important systemic review highlighting the HCC risk in NAFLD (Clin Gastroenterol Hepatol. 2012;10:1342-1359.e2.) should be cited in ‘Discussion’.

Re: following reviewer’s comment, we cited this important review in our discussion.

  • There are some minor typographical errors that should be corrected (abbreviations, points, spaces, grammar-related, mellitus in cursive …).

Re: Thank you for your observation. Typographical errors have been corrected.